# Symptom trajectories of non-cancer patients in the last six months of life: Identifying needs in a population-based home care cohort

**Katrin Conen** [ID][1,2]\*, **Dawn M. Guthrie**[3,4], **Tara Stevens**[3], **Samantha Winemaker**[2], **Hsien Seow** [ID][2,5]

**1** Department of Medicine, McMaster University, Hamilton, Ontario, Canada, **2** Department of Family Medicine, Division of Palliative Care, McMaster University, Hamilton, Ontario, Canada, **3** Department of Kinesiology & Physical Education Wilfrid Laurier University, Waterloo, Ontario, Canada, **4** Department of Health Sciences, Wilfrid Laurier University, Waterloo, Ontario, Canada, **5** Department of Oncology, McMaster University, Hamilton, Ontario, Canada

\* conenk@mcmaster.ca

## Abstract

### Introduction

The end-of-life symptom prevalence of non-cancer patients have been described mostly in hospital and institutional settings. This study aims to describe the average symptom trajectories among non-cancer patients who are community-dwelling and used home care services at the end of life.

### Materials and methods

This is a retrospective, population-based cohort study of non-cancer patients who used home care services in the last 6 months of life in Ontario, Canada, between 2007 and 2014. We linked the Resident Assessment Instrument for Home Care (RAI-HC) (standardized home care assessment tool) and the Discharge Abstract Databases (for hospital deaths). Patients were grouped into four non-cancer disease groups: cardiovascular, neurological, respiratory, and renal (not mutually exclusive). Our outcomes were the average prevalence of these outcomes, each week, across the last 6 months of life: uncontrolled moderate-severe pain as per the Pain Scale, presence of shortness of breath, mild-severe cognitive impairment as per the Cognitive Performance Scale, and presence of caregiver distress. We conducted a multivariate logistic regression to identify factors associated with having each outcome respectively, in the last 6 months.

### Results

A total of 20,773 non-cancer patient were included in our study, which were analyzed by disease groups: cardiovascular (n = 12,923); neurological (n = 6,935); respiratory (n = 6,357); and renal (n = 3,062). Roughly 80% of patients were > 75 years and half were female. In the last 6 months of life, moderate to severe pain was frequent in the cardiovascular (57.2%), neurological (42.7%), renal (61.0%) and respiratory (58.3%) patients. Patients with renal

---

**Data Availability Statement:** Data are available from the Canadian Institute for Health Information for researchers who meet the criteria for access to

confidential data. Interested readers can access these data in the same manner as the authors. These data represent third party data that are not owned nor collected by the study authors. A data request form can be found here: https://www.cihi.ca/en/access-data-and-reports/make-a-data-request.

**Funding:** This work is funded by the Canadian Centre for Applied Research in Cancer Control (ARCC). ARCC receives core funding from the Canadian Cancer Society Research Institute (grant #2015-703549). The senior author is also supported by the Canada Research Chairs program. Authors otherwise did not receive funding for this work.

**Competing interests:** The authors do not disclose any competing interests for this submission.

disease had significantly higher odds for reporting uncontrolled moderate to severe pain (odds ratio [OR] = 1.21; 95% CI: 1.08 to 1.34) than those who did not. Patients with respiratory disease reported significantly higher odds for shortness of breath (5.37; 95% CI, 5.00 to 5.80) versus those who did not. Patients with neurological disease compared to those without were 9.65 times more likely to experience impaired cognitive performance and had 56% higher odds of caregiver distress (OR = 1.56; 95% CI: 1.43 to 1.71).

## Discussion

In our cohort of non-cancer patients dying in the community, pain, shortness of breath, impaired cognitive function and caregiver distress are important symptoms to manage near the end of life even in non-institutional settings.

## Introduction

Multiple randomized controlled trials, and other clinical trials, have shown that a palliative approach to care is beneficial to improve the dying experience and patient outcomes including improved well-being, symptom management, quality-of-life, satisfaction with care and decreased caregiver distress and Emergency Department visits at the end of life [1–8]. Despite evidence of the benefits of palliative care in non-cancer populations, referrals to palliative care services are more often happening in cancer patients versus non-cancer patients [9–14]. One reason for this might be that unmet symptoms and their symptom trajectories are very well described in the cancer population, as compared to the non-cancer population (e.g., chronic lung disease, chronic heart disease, renal disease and Alzheimer's dementia) where the illness trajectory tends to be much less predictable [15–21].

Research shows that patients with advanced cancer diagnoses compared to non-cancer diagnoses (e.g. chronic obstructive pulmonary disease, congestive heart failure, etc.) have similar needs at the end-of life, including needs for emotional well-being, physical functioning and quality of life [22]. However, non-cancer patients often receive palliative care supports later in the illness trajectory. For instance, in retrospective studies of cancer vs. non-cancer patients, non-cancer patients presented with lower functional status when initially referred to palliative care [23, 24]. A systematic review of 15 studies around end-of-life needs of non-cancer patients reported the body of research-to-date as being qualitative and descriptive and suggested more longitudinal and observational studies are necessary to identify patients that would benefit from a palliative care referral in the context of their illness and associated symptom trajectories [16]. Many of these studies were conducted in hospital and institutional settings, and thus population-based symptom prevalence over time, particularly for community-dwelling patients dying at home, has not been well-studied.

To address this gap, our study focuses on a population-based non–cancer cohort in Ontario, Canada who accessed publicly-funded home care services. All individuals using homecare service receive a standardized assessment, specifically the Resident Assessment Instrument for Home Care (RAI-HC). The RAI-HC is completed every six months, providing us with a large sample of diverse patients who are followed over time in the community until death. Our study aimed to describe the average symptom trajectories for a cohort of non-cancer patients in the last six months of life and identify factors associated with having a symptom issue.

## Materials and methods

### Study design, participants, and setting

This is a retrospective, population-based cohort study of non-cancer patients who accessed publicly-funded home care services in the province of Ontario, Canada between January 1, 2007 to March 31, 2014. To be included, patients had to have a documented death during the study period (either at home or hospital), have used home care (and thus have a home care assessment) in the final six months of life, and a non-cancer diagnosis (as per the home care assessment).

### Data sources

We used routinely collected clinical health administrative data. Specifically, our study merged the Resident Assessment Instrument for Home Care (RAI-HC) database for home care and the Discharge Abstract Database (DAD) for hospitals at the individual-level through unique health insurance numbers. (See S1 Fig). Individuals expected to receive at least 60 days of home care and receive a standardized assessment tool, called the RAI-HC (akin to the Minimum Data Set in the USA). This assessment tool is mandated by the province for billing, accountability, and research purposes. The RAI-HC is completed in the patient's home by a trained professional (typically a registered nurse) on a laptop, following a detailed coding manual [25]. Thus, the tool contains provider-reported outcome measures. The assessment is repeated roughly every 6 months, unless there is a major change in health status or a discharge from hospital [26]; thus patients can have multiple assessments completed. The assessor completes the RAI-HC based on an interview with the patient and their family in their homes and using their best clinical judgement. The assessment includes, but is not limited to, items that measure the client's functional status, psychosocial well-being, physical health, and care needs [27]. There have been multiple studies that attest to the reliability and validity of items within the RAI-HC [25, 28–30]. If a patient dies while receiving home care, date of death is document in the RAI-HC. If a patient dies in hospital, date of death is recorded in the Discharge Abstract Database (DAD).

### Variables

Our main variable was non-cancer diagnosis. Patients in the study population were grouped into four non-mutually exclusive diagnostic categories: 1) cardiovascular (cerebrovascular accident, congestive heart failure, coronary artery disease, peripheral vascular disease); 2) neurological (Alzheimer's dementia, dementia [other than Alzheimer's], multiple sclerosis, parkinsonism); 3) respiratory (emphysema, chronic obstructive pulmonary disease, asthma); and 4) renal failure as indicated on the RAI-HC assessment (item J1a-ac). If a patient had cancer (item J1x), they were excluded from the cohort. Disease groups are not mutually exclusive, since individuals often have multiple co-morbid chronic conditions. To compare patients equally over time, we aligned patients' date of death as time zero and then counted backwards 26 weeks (approximately 6 months) from death.

### Outcomes

All outcomes of interest were derived from the RAI-HC assessment and included pain, shortness of breath (physical symptoms), cognitive performance and caregiver distress (psychosocial symptoms).

1. *Pain*: Moderate-severe daily pain that was also uncontrolled was measured by having a score of 2 or higher out of 4 on the Pain Scale (item K4a-b) (meaning the frequency is daily

and the intensity is moderate to severe) and that the pain was uncontrolled (item K4e) (i.e., "medications do not adequately control pain") [31].

2. *Shortness of breath* (item K3e): "Shortness of breath was present in the past 3 days" (yes/no)

3. *Cognitive performance*: Mild-severe cognitive impairment was measured as a score of 2 or higher out of 6 on the Cognitive Performance scale (CPS) (item B1-2 and C) [32]. The Cognitive Performance scale is a hierarchical screener which includes two items found on traditional cognitive assessments (e.g., short-term memory, daily decision making) and two items reflecting functional status (e.g., expressive communication, independence in eating). The scale ranges from zero to six (0 = no cognitive impairment; 1 = borderline intact; 2 = mild impairment; 3 = moderate impairment; up to 6 = very severe impairment).

4. *Caregiver distress* (item G2c): "Patient's primary informal caregiver experiences feelings of anger, depression or distress" (yes/no).

## Covariates

Other dichotomous covariates included: i) caregiver lives with patient (item G1e) (yes/no); ii) death in hospital (yes/no); iii) loss of appetite (item K2d) (yes/no); iv) social decline causing distress (item F2) (yes/no); v) signs and symptoms of depression as measured by the Depression Rating Scale (DRS) [33] score of 3 or more (item E1-4) (yes/no); and vi) moderate-severe impairment as measured by the Activities of Daily living (ADL) Self-performance Hierarchy scale [34] score of 2 or more (item H1-7) (yes/no). These covariates were shown to be associated with the outcomes in prior research [35].

## Statistical methods

We used data from all RAI-HC assessments in any patient's last 26 weeks of life to create the average trajectory of each symptom over time. When describing the demographic and health characteristics of our cohort, only the most recent RAI-HC assessment for each individual was used. The data present the proportion of patients who completed a RAI-HC from 26 weeks until one week (which represented 0–7 days) before death, and who had that symptom/issue present. Multivariate logistic regression models were created to compare the odds of having the outcomes respectively in the final 6 months of life, controlling for age, sex, disease group, and other covariates described above. All results were reported as an adjusted odds ratio (OR) with 95% confidence interval (CI) and a two-tailed alpha level of 0.05 was used to define statistical significance. As a sensitivity test, we examined the outcomes by those who died in hospital versus died at home separately; this was to explore the potential for selection bias, whereby patients who were more symptomatic would be admitted to hospital before reporting symptoms in the home care assessment. Analyses were conducted using SAS version 9.4. The study was approved and deemed exempt by Hamilton Integrated Research Ethics Board (Project #3039) and the Wilfrid Laurier University Research Ethics Board (REB #5310) as it used de-identified secondary data analysis. All necessary permissions and approval to access the data were obtained from the Canadian Institute for Health Information (CIHI).

## Results

In our study population of home care patients assessed between 2007–2014, the total number of unique individuals that contributed assessments during the last six months of life and fit the study criteria was 37,981. After excluding individuals with a cancer diagnosis from this group,

**Table 1. Characteristics and overall symptom burden by disease group.**

| | Cardiovascular (n = 12 923) | Neurological (n = 6935) | Respiratory (n = 6357) | Renal (n = 3062) |
|---|---|---|---|---|
| | % (n) | | | |
| **Age** | | | | |
| Under 65 | 6.1 (782) | 3.4 (237) | 7.9 (499) | 10.8 (332) |
| 65–74 | 12.7 (1642) | 8.7 (605) | 18.0 (1142) | 16.4 (503) |
| 75–84 | 34.4 (4445) | 35.9 (2492) | 37.4 (2379) | 36.1 (1104) |
| 85+ | 46.8 (6052) | 51.9 (3601) | 36.8 (2337) | 36.7 (1123) |
| **Sex** | | | | |
| Male | 48.5 (6267) | 47.0 (3257) | 48.5 (3083) | 53.4 (1634) |
| Female | 51.5 (6654) | 53.0 (3678) | 51.5 (3083) | 46.6 (1428) |
| **Marital Status** | | | | |
| Married | 43.3 (5591) | 48.2 (3343) | 41.7 (2649) | 49.1 (1503) |
| Primary caregiver lives with client | 57.9 (7477) | 63.5 (4403) | 55.8 (3550) | 63.4 (1942) |
| **Education** | | | | |
| Completed Gr. 11 or less | 62.2 (8043) | 60.2 (4177) | 64.1 (4077) | 63.3 (1937) |
| Completed college, university or trade school | 21.6 (2792) | 23.4 (1625) | 18.9 (1199) | 21.4 (656) |
| **Patient factors** | | | | |
| Signs/symptoms of depression (DRS score of > = 3) | 21.3 (2756) | 23.8 (1650) | 22.6 (1438) | 22.0 (672) |
| Moderate to severe impairment in activities of daily living (ADL) (rates 2 and up) | 31.0 (4006) | 50.2 (3483) | 23.4 (1487) | 29.7 (908) |
| Decline in social activities that causes the person distress | 15.1 (1948) | 7.8 (539) | 16.6 (1055) | 16.7 (511) |
| **Outcome measures** | | | | |
| Moderate to severe pain (Pain Scale score > = 2) | 57.2 (7392) | 42.7 (2961) | 58.3 (3708) | 61.0 (1868) |
| Mild to severe cognitive impairment (CPS score of > = 2) | 54.4 (7026) | 91.3 (6330) | 45.2 (2873) | 50.8 (1554) |
| Caregiver experiences feelings of anger, distress or depression | 26.7 (3447) | 37.7 (2612) | 23.9 (1516) | 28.1 (861) |
| **Timing of patient's closest assessment to death** | | | | |
| 0–4 weeks before death | 20.7 (2676) | 19.3 (1338) | 20.9 (1331) | 21.6 (662) |
| 5–12 weeks before death | 37.2 (4808) | 36.6 (2541) | 37.9 (2410) | 36.2 (1108) |
| 13–26 weeks before death | 42.1 (5439) | 44.1 (3056) | 41.2 (2616) | 42.2 (1292) |

the final sample size of unique individuals was 20,773 (33,596 assessments). Based on non-exclusive diagnosis categories, we had home care patients grouped into cardiovascular (n = 12,923), neurological (n = 6,935), respiratory (n = 6,357) and renal (n = 3,062) diagnoses. Overall, 64% patients died in the hospital. In our cohort, 42.4% had their most recent home care assessment 3 to 6 months before death, 36.9% in the 1–3 months before death, and 20.6% in the final 1 month of life.

Most of the patients were over the age of 75 years old (ranging from 81.2% in the circulatory, 87.8% in the neurological, 74.2% in the respiratory and 72.8% in the renal group). Half of the population were female. Approximately 60% of patients lived with a primary caregiver (Table 1). One-fifth of patients showed signs and symptoms of depression across the four disease groups. Moderate to severe impairment in completing Activities of Daily Living were highest in the neurological group (50.2%) compared to 31.0% in the circulatory, 23.4% in the respiratory, and 29.7% in the renal group. Social decline that caused distress was found in approximately 15% of patients in the disease groups, though was lower in the neurological group (7.8%).

Examining outcomes in the last assessment closest to death, there was a higher prevalence of moderate-severe pain in the cardiovascular (57.2%), renal (61.0%) and respiratory group

(a)

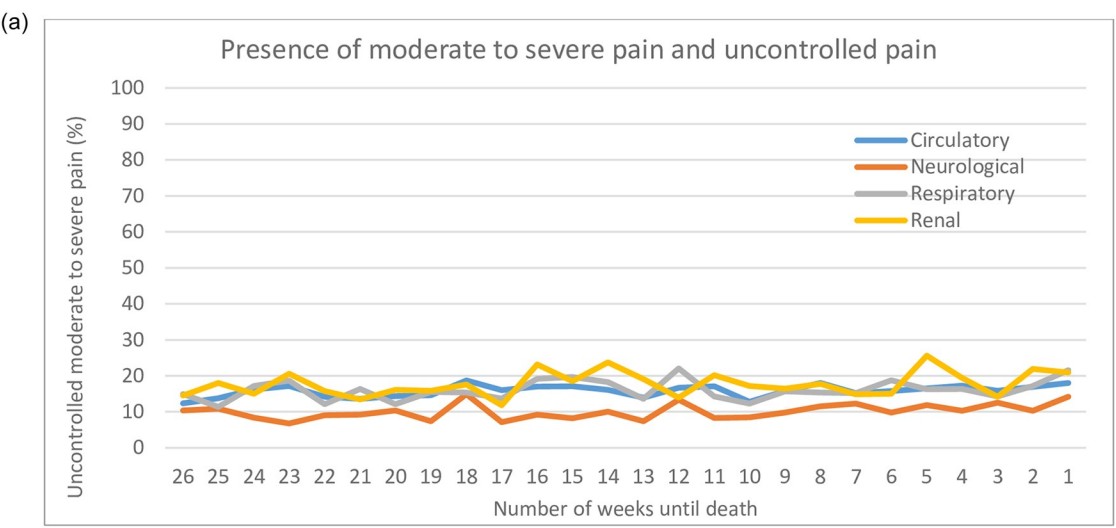

(b)

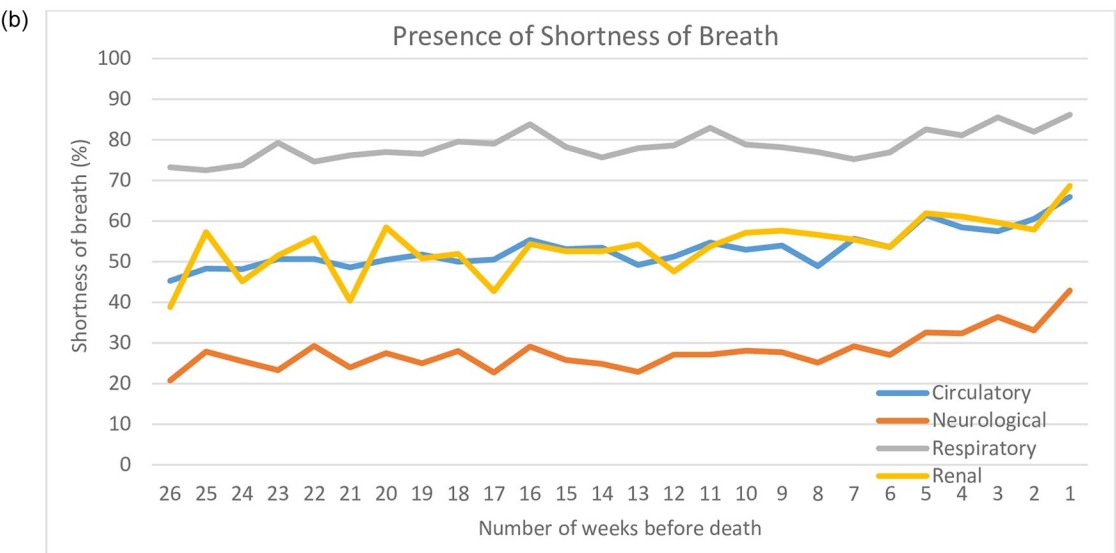

**Fig 1. Physical symptom prevalence in the last 6 months of life.**

(58.3%), compared to the neurological group (42.7%) (Table 1). 91.3% of patients with neurological disease had documented mild-severe cognitive impairment. Shortness of breath was reported in 70–85% of patients grouped in the respiratory category. This was on average lower reported in the circulatory, renal, and neurological group (40–65%, 45–65% and 20–40%, respectively).

Mean symptom trajectories over the last 26 weeks of life across the 4 disease groups are shown in Figs 1 and 2. Overall, there was a consistent proportion of patients reporting symptoms prevalence each week across the last 6 months of life for uncontrolled moderate-severe pain, mild-severe cognitive impairment, and caregiver distress; in fact, the prevalence for these symptoms increased slightly by 5–10% closer to death. While moderate to severe pain was reported in nearly half of disease groups, the proportion who also rated that pain as uncontrolled pain dropped to approximately 20% of patients across all disease groups. Cognitive impairment was consistently prevalent in nearly half the disease groups, with the exception

(a)

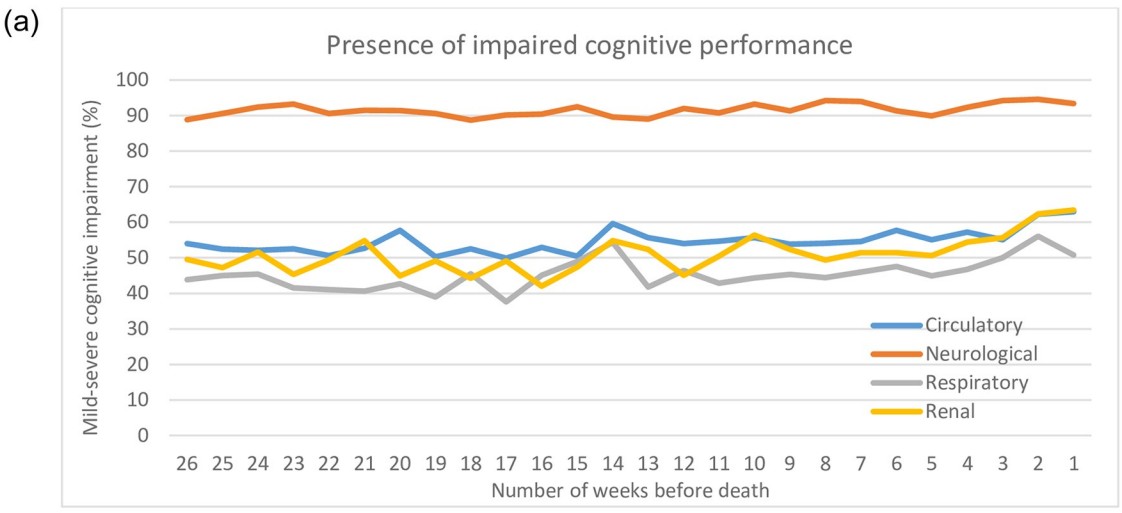

(b)

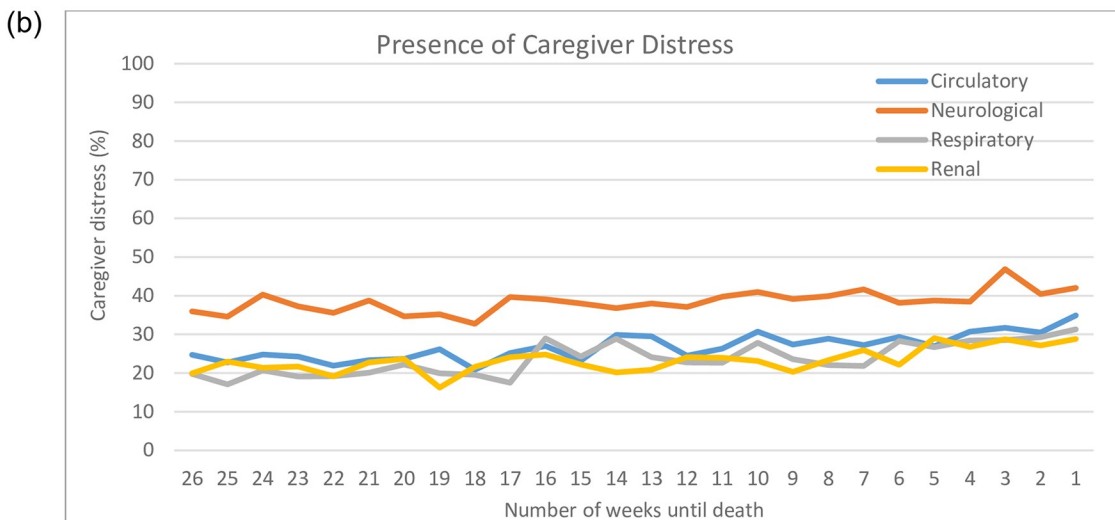

**Fig 2. Psychosocial symptom prevalence in the last 6 months of life.**

being those with neurological disease, where it remained at higher than 90%. Caregiver distress was also evident in about 20% of patients and this proportion rose by 10% or more as one approached death; note neurological disease groups had higher rates over time starting at 35% 6 months before death.

There was more variation in the trajectory of prevalence of shortness of breath. Those with respiratory disease had the highest average prevalence, beginning with a proportion of 73% in the 6 months before death, which rose to 86% in the final week of life. Those with cardiovascular or renal diseases began with roughly 42% reporting shortness of breath, which rose to a prevalence of 69% in the week before death. Neurological disease began at 21% and rose to 43% over the last 6 months of life. Nonetheless, regardless of the disease groups, the prevalence of shortness of breath increased roughly 15%-20% or more in the last four weeks of life. Our sensitivity analysis showed that there was no difference in the symptom trajectories among those who died in hospital vs. died at home across the 4 disease groups.

**Table 2. Adjusted odds ratio of having symptoms (moderate to severe pain, shortness of breath, cognitive impairment, caregiver distress, self-reported poor health) in the last six month of life using multivariate logistic regression analysis controlling for covariates\*.**

| | | Moderate to severe pain and uncontrolled | | Shortness of breath | | Mild to severe cognitive impairment | | Caregiver distress | |
|---|---|---|---|---|---|---|---|---|---|
| | | Odds ratio (95% confidence interval) | | | | | | | |
| Age (reference: <65) | 65–74 | **0.69** | **(0.59 to 0.82)** | **1.29** | **(1.11 to 1.50)** | **1.35** | **(1.15 to 1.57)** | 1.13 | (0.69 to 1.33) |
| | 75–84 | **0.62** | **(0.53 to 0.71)** | **1.38** | **(1.21 to 1.58)** | **1.93** | **(1.67 to 2.22)** | **1.22** | **(1.05 to 1.41)** |
| | ≥85 | **0.60** | **(0.51 to 0.69)** | **1.41** | **(1.23 to 1.61)** | **3.16** | **(2.74 to 3.64)** | **1.28** | **(1.10 to 1.48)** |
| Sex (reference: male) | Female | **1.24** | **(1.15 to 1.35)** | 0.97 | (0.91 to 1.03) | 0.94 | (0.88 to 1.01) | **0.76** | **(0.71 to 0.81)** |
| Cardiovascular Diagnosis (reference: no) | Yes | 0.82 | (0.13 to 1.36) | **1.39** | **(1.29 to 1.50)** | **1.14** | **(1.04 to 1.23)** | **1.28** | **(1.05 to 1.22)** |
| Neurological Diagnosis (reference: no) | Yes | 1.01 | (0.73 to 1.92) | **0.53** | **(0.49 to 0.58)** | **9.65** | **(8.67 to 10.73)** | **1.56** | **(1.43 to 1.71)** |
| Respiratory Diagnosis (reference: no) | Yes | **1.77** | **(1.61 to 1.96)** | **5.37** | **(5.00 to 5.80)** | 0.92 | (0.85 to 1.00) | 0.95 | (0.87 to 1.03) |
| Renal Diagnosis (reference: no) | Yes | **1.21** | **(1.08 to 1.34)** | **1.18** | **(1.08 to 1.28)** | 1.05 | (0.96 to 1.15) | **1.13** | **(1.02 to 1.24)** |
| Died in hospital (reference: died at home) | Yes | **1.11** | **(1.02 to 1.21)** | 0.97 | (0.91 to 1.06) | **0.76** | **(0.71 to 0.82)** | 1.00 | (0.94 to 1.08) |

\* Each of the four models was adjusted for these additional covariates: caregiver lives with patient; moderate-severe impairment in Activities of Daily Living; social decline causing distress; signs and symptoms of depression; and loss of appetite.

\*\* bold indicates statistical significance (p<0.05).

Table 2 shows the results of the multivariable logistic regression on the factors associated with having the outcomes in the last six months of life. During the last six months of life, age did not consistently affect the odds of reporting symptom scores. Older age increased the likelihood of experiencing shortness of breath (OR: 1.29 to 1.41), impaired cognitive performance (OR: 1.35 to 3.16) and caregiver distress (OR: 1.13 to 1.28). Females had significantly higher odds for reporting uncontrolled pain (OR: 1.24; 95% CI, 1.15 to 1.35). Those with neurological disease had higher odds for impaired cognitive performance (9.65; 95% CI, 8.67 to 10.73) and caregiver distress (1.56; 95% CI, 1.43 to 1.71) than those without neurological disease. Those with respiratory disease reported significantly higher odds for shortness of breath (5.37; 95% CI, 5.00 to 5.80) and uncontrolled pain (1.77; 95% CI, 1.61 to 1.96) than those without respiratory disease. Cardiovascular patients reported significantly higher odds for shortness of breath (1.39; 95% CI, 1.29 to 1.50), impaired cognitive performance (1.14; 95% CI, 1.04 to 1.23), and caregiver distress (1.28; 95% CI, 1.05 to 1.22) compared to those without cardiovascular disease. Patients with renal disease reported significant higher odds for pain (1.21; 95% CI, 1.08 to 1.34), shortness of breath (1.18; 95% CI, 1.08 to 1.28) and caregiver distress (1.13; 95% CI 1.02 to 1.24) compared to those without renal disease. Those who died in hospital were more likely to have uncontrolled moderate-severe pain (1.11; 95% CI, 1.02 to 1.21) and less likely to have cognitive impairment (0.76; 95% CI, 0.71 to 0.82).

## Discussion

Our data present trajectories of symptoms in the last six months of life in a non-cancer population of home care patients among four disease groups: cardiovascular, neurological, renal, and respiratory. Across all non-cancer disease groups, the trajectory of symptom prevalence increased slightly each week towards death. Cognitive impairment was evident in at least half of the patients in the disease groups, and over 90% in the neurological group. Prevalence of shortness of breath rose by 20% over time across all groups, with the highest prevalence being among those with respiratory disease at 86% in the last week of life. Caregiver distress rose by 10% over time and was prevalent in 35%-40% of patients in the final weeks of life. With a sample size of 20,773 assessments, this is a very large population-based cohort focusing on describing average weekly symptom prevalence among those receiving home care.

Pain, a leading symptom and concern in cancer patients at the end of life [19, 36, 37], was reported as moderate to severe in nearly half or more of the non-cancer cohort, yet only one-fifth described the pain as uncontrolled. This suggests pain may be well-managed by home care services, and pain intensity alone is insufficient to understand one's overall pain experience. Having renal disease and respiratory disease, respectively, compared to not having those disease, increased one's odds for moderate to severe uncontrolled pain. Reasons for this are likely complex and multifactorial. Patients with renal and respiratory diseases might not receive enough narcotic treatments, as there are reported concerns in starting higher narcotic treatment strategies in chronic respiratory and renal diseases based on concerns around respiratory depression [36, 38, 39]. Additionally, shortness of breath is known to become more common and severe in the final stage of patients with cancer and chronic obstructive pulmonary disease [40]. In our analysis, shortness of breath increased in prevalence across all four disease groups in the final 6 months of life. Patients with respiratory, renal and cardiovascular diseases reported higher prevalence of shortness of breath, in line with other literature [41, 42].

As expected, patients with neurological disease had the highest prevalence of cognitive impairment and caregiver distress among the four groups. This finding supports prior literature linking caregiver distress with caring for a relative with cognitive impairment [43–47]. Caregivers perform a critical role in the socioeconomic context of providing care to a dying patient. To maintain sustainability of this form of care, caregiver needs must be identified, and support systems must be made available accordingly. Ultimately, understanding the trajectory of symptoms and the factors that are associated with increased odds of having complex symptoms can help to identify earlier those who could benefit from palliative care services. This includes non-cancer patients dying at home, where multidisciplinary treatment approaches such as physiotherapy, psychosocial support and better symptom management can improve symptom burden and patient and family outcomes.

Using administrative home care data to describe the weekly average symptom prevalence in the 6 months before death has limitations and strengths. The limitations include the real potential for selection bias in that we lose out on data from patients with very complex symptom issues who then refuse home care services or when they go to hospital; thus, the symptoms of each disease group at those points could be under-reported. We did examine those who died in hospital compared to those who died at home as a sensitivity test, and found no difference in the symptom trajectories, though those dying in hospital were more likely to have uncontrolled pain. Also other data show most terminal hospitalizations are less than 2 weeks and home care is protective of end-of-life hospitalizations [48]. Moreover, it is also possible that those with very complex symptoms would be more willing to accept home care services. Nonetheless, the timing of these formal assessments are typically far apart and only about half the patients had repeated measures, meaning that the trajectories are an average of the cohort and not individual trajectories of symptoms reported weekly. However, a strength of our approach is that it avoids some of the major issues with conducting studies at end of life, which include low recruitment, high missing data, and high attrition rates because patients are too tired or sick to participate [49]. Also in our study, there is virtually no missing data, as the RAI-HC is a mandatory standardized clinical assessment for most individuals receiving publicly-funded home care. Thus, our data is an inclusive population-based cohort, producing a large sample size, and allows us to look at trajectories over time on a weekly basis (for the subset of the cohort who reported in that week).

Other limitations of our data are the inability to have mutually exclusive data for our four analyzed disease groups and control for specific comorbidities. This should be addressed in subsequent research with broader data linkage. Some outcomes, such as shortness of breath, were dichotomous, and do not capture intensity as other validated measures do [50]. Our

study is not able to describe the quality of care nor details around symptom management. Since the RAI-HC does not define whether or not the person received specialized palliative care, it is unclear whether changes in treatment plans or initiations of other supportive measures were initiated; this could be addressed in future research. Focusing on users of publicly-funded home care at the end of life means we do not have data on those who did not use home care services, strictly used private home care services, or died in long-term care (approximately 20–25% of the population).

## Conclusions

In conclusion, our study describes symptom trajectories in non-cancer home care recipients in Ontario, Canada at end of life. We found across all non-cancer disease groups; the trajectory of symptom prevalence increased slightly each week towards death. Moderate to severe pain was prevalent in nearly half or more of the cohort, but only one-fifth described the pain as uncontrolled. In contrast, shortness of breath, impaired cognitive function and caregiver distress were more highly and consistently prevalent across time near the end of life. Our results suggest the non-cancer population has unmet symptoms needs outside institutional settings.

## Supporting information

**S1 Fig. CONSORT diagram.**
(DOCX)

**S2 Fig. The RECORD statement.**
(DOCX)

## Author Contributions

**Conceptualization:** Katrin Conen, Dawn M. Guthrie, Samantha Winemaker, Hsien Seow.

**Data curation:** Dawn M. Guthrie.

**Formal analysis:** Tara Stevens.

**Funding acquisition:** Hsien Seow.

**Methodology:** Hsien Seow.

**Project administration:** Hsien Seow.

**Supervision:** Dawn M. Guthrie, Hsien Seow.

**Validation:** Hsien Seow.

**Writing – original draft:** Katrin Conen.

**Writing – review & editing:** Katrin Conen, Dawn M. Guthrie, Samantha Winemaker, Hsien Seow.

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
