## [Decision Letter · Decision Letter 0]

18 Mar 2021

PONE-D-21-00638

Symptom trajectories of non-cancer patients in the last six months of life: Identifying needs for palliative care

PLOS ONE

Dear Dr. Katrin Cohen, 

Thank you for submitting your manuscript to PLOS ONE. After careful consideration, we feel that it has merit but does not fully meet PLOS ONE’s publication criteria as it currently stands. Therefore, we invite you to submit a revised version of the manuscript that addresses the points raised during the review process.

ACADEMIC EDITOR:

Thank you for submitting this manuscript to PLOS ONE. It is an important study. The peer reviewers have given detailed consideration of your manuscript. See points below. Please review and respond to the editor and peer review comments below.

Two main issues that need to be addressed.

1. Reporting of the methods - please use reporting guidance for observational studies and extension for routine data. This is STROBE and extension RECORD. Please review these guidance, complete for your manuscript, indicate in your methods reporting using this guidance and reference, and the completed checklist included as a supplementary file.

2. The manuscript is comparing assessments from different patients conducted at different times prior to death. This is a limitation compared to a prospective longitudinal study, as the individual trajectories cannot be seen. This should be discussed and indication of a limitation on the study. 

We look forward to receiving your revised manuscript.

Kind regards,

Catherine J Evans, PhD, MSc, BSc (Hons)

Academic Editor

PLOS ONE

Additional Editor Comments (if provided):

See above

Journal Requirements:

2)  We note that you have indicated that data from this study are available upon request. PLOS only allows data to be available upon request if there are legal or ethical restrictions on sharing data publicly. For information on unacceptable data access restrictions, please see http://journals.plos.org/plosone/s/data-availability#loc-unacceptable-data-access-restrictions.

3) We note that you have included the phrase “data not shown” in your manuscript. Unfortunately, this does not meet our data sharing requirements. PLOS does not permit references to inaccessible data. We require that authors provide all relevant data within the paper, Supporting Information files, or in an acceptable, public repository. Please add a citation to support this phrase or upload the data that corresponds with these findings to a stable repository (such as Figshare or Dryad) and provide and URLs, DOIs, or accession numbers that may be used to access these data. Or, if the data are not a core part of the research being presented in your study, we ask that you remove the phrase that refers to these data.

4) Please include your tables as part of your main manuscript and remove the individual files. Please note that supplementary tables (should remain/ be uploaded) as separate "supporting information" files

5) PLOS requires an ORCID iD for the corresponding author in Editorial Manager on papers submitted after December 6th, 2016. Please ensure that you have an ORCID iD and that it is validated in Editorial Manager. To do this, go to ‘Update my Information’ (in the upper left-hand corner of the main menu), and click on the Fetch/Validate link next to the ORCID field. This will take you to the ORCID site and allow you to create a new iD or authenticate a pre-existing iD in Editorial Manager. Please see the following video for instructions on linking an ORCID iD to your Editorial Manager account: https://www.youtube.com/watch?v=_xcclfuvtxQ

Reviewers' comments:

Reviewer's Responses to Questions

**Comments to the Author**

1. Is the manuscript technically sound, and do the data support the conclusions?

Reviewer #1: No

Reviewer #2: Partly

Reviewer #3: No

2. Has the statistical analysis been performed appropriately and rigorously? 

Reviewer #1: Yes

Reviewer #2: Yes

Reviewer #3: No

3. Have the authors made all data underlying the findings in their manuscript fully available?

Reviewer #1: Yes

Reviewer #2: Yes

Reviewer #3: No

4. Is the manuscript presented in an intelligible fashion and written in standard English?

Reviewer #1: Yes

Reviewer #2: Yes

Reviewer #3: Yes

5. Review Comments to the Author

Reviewer #1: Thank you for submitting this manuscript. This large observational study on symptom trajectories of non-cancer patients in the last six months of life addresses an important topic and is generally well written. However, I have some concerns about the methodological approach and the reporting.

Major concerns:

1. The 6 month symptom trajectories are constructed from single individual symptom scores with varying time until death, rather than from individual repeated symptom measures. A potential issue with this approach is that it is very plausible that those who are close to death and more symptomatic may be more likely to decline a study visit and are missing from the sample (selection bias). This potential source of bias brings some doubt to the conclusion that symptom trajectories towards the end of life tend to be flat.

2. The aim of the study is not clear and does not match with the study described. “Our study aimed to identify gaps in knowledge among patients with a variety of serious illnesses that would benefit from a palliative approach to care.” It is not clear to me how this study identifies knowledge gaps. Please consider what this study aims to achieve and its contribution to the literature.

Minor points:

Methods

• Please clarify how the data is collected when the subject has cognitive impairment e.g. for self-reported health?

• “Shortness of breath was measured by asking: Do you feel short or breath, yes or no? Answers were documented based on the assessor’s judgement after the interview with the patient.” These two sentences appear contradictory. Please clarify under what circumstances the assessor would override a patient’s assessment of their shortness of breath?

• The information provided about pain measurement is unclear. Is the main outcome the combination of moderate/severe AND uncontrolled? Please revise so that the methods are clear and reproducible.

• The grouping for psychosocial symptoms seems unusual - does self-reported health fit here?

• Please justify why you selected the 5 symptoms as the main outcomes.

• There is a wide timeframe within which RAI assessments can take place (6-12 months). Please can you detail reasons for this and the implications on the sample? For instance, could patients who are feeling more unwell decline the visits? Please can you provide some critique and discussion around the approach and possibility of bias.

Results

• Table 1, what does bold indicate? Please detail in the footnotes.

• In Table 2, please detail exactly which covariates are included in each model.

Discussion:

• In discussion: “Based on our descriptive and multivariate analyses, we could demonstrate that patients with non-cancer overall seem not to suffer severely from symptom needs in their disease trajectory over the last six months of life.” What is this based on? I don’t think I would conclude the same based on the data presented in the figures.

• Hypothesis referred to in discussion does not correspond to the hypothesis detailed in the background. Please address this discrepancy.

• The trajectories presented are hypothetical and do not relate to individual trajectories. Please discuss the possible implications and the limitations of this approach.

• Overall the manuscript would benefit from adhering to reporting guidelines, e.g. STROBE for observational studies.

Reviewer #2: Thanks for the opportunity to review this manuscript, which uses a robust regional routinely collected clinical dataset to retrospectively investigate symptom trajectories of people who died from non-cancer illnesses. I think the approach taken is appropriate, however there are several areas in which the reporting of the methods could be improved or clarification is needed, therefore I think revisions are required before publication. I have made comments below to suggest how this might be done.

General comments

1. I think the methods section could be organised more clearly to aid the reader (see specific comments below.) I would also recommend that the authors use a checklist for reporting of this type of study e.g. the RECORD statement

https://journals.plos.org/plosmedicine/article?id=10.1371/journal.pmed.1001885

2. Focusing on deaths at home/in hospital only means that the study is unable to comment on those who died in a care home or hospice (what proportion of deaths is this in Ontario?). These two groups might have very different symptom burden and so this is a limitation of the study which should be discussed

3. As I understand it the study is looking at ‘average trajectories’ i.e. rather than looking at the change in each individual’s symptoms over 6 months, the authors are comparing assessments from different patients conducted at different times prior to death. This is a limitation compared to a prospective longitudinal study, as the individual trajectories cannot be seen. This should be discussed.

4. Given the aim, I'm not sure how the multivariable analysis adds to this study. It doesn’t add to the analysis of symptom trajectories, since it only gives the odds of having each outcome at any point in the last 6 months of life dependent on the characteristic, not the change over time. Please could the authors more clearly justify why they conducted this analysis and how it contributes to their aim.

Specific comments

Abstract

1. Please state the data source explicitly in the abstract. E.g. ‘retrospective study using data from the Canadian institute for health.’

2. The aim in the abstract does not match that in the main paper. I think that the aim is probably to analyse symptom trajectories and identify differences, rather than specifically to identify gaps in knowledge? Please could you clarify and ensure consistency between abstract and main paper.

3. “Patients were grouped into four non-cancer disease groups such as”. There is no need for “such as”, all groups are described

4. When reporting odds in the abstract, please state the comparator. E.g. 'renal patients had higher odds of pain compared to other groups' etc.

5. “symptom trajectories vary with disease group”. Do the trajectories differ, or is it symptom prevalence that varies?

Introduction

6. Introduction line 4 “satisfactory” – do you mean satisfaction?

7. 3rd sentence. How does the possibility that palliative care referrals are often made for symptom management explain the reduced referrals in non-cancer diagnoses? Symptoms are known to be high in non-cancer too (as the authors discuss later on). Please rephrase to clarify the argument

8. Please move the description of the frequency with which the RAI-HC is completed to the methods section.

9. I think the aim in the introduction is clear, but I’m a bit confused by the hypothesis: why do the authors hypothesise different symptom patterns in different non cancer illnesses? What existing evidence has led them to this hypothesis? Also, by “gaps of knowledge” do the authors mean differences in symptom patterns which would therefore allow a more nuanced approach to palliative care referral? Please clarify.

Methods

10. At the start of the methods, please state that you are using routinely collected clinical data.

11. The RAI-HC may not be familiar to international readers. It would be helpful if it could be introduced and described in a single section. At the moment the description is spread across the introduction and several sections of the methods. Perhaps this information can be combined into a single description of what the RAI-HC is, how it is completed & how it was used here

12. Re: diagnostic categories, were there no deaths with liver failure? Or were these combined into another category

13. Last sentence of ‘population’ section. I think this would fit better at the start of the 'analysis' section.

14. Whilst the pain outcome is detailed, the shortness of breath outcome is a yes/no question. I recognise the authors are limited by the dataset, but could they comment on the effect on symptom prevalence of using this measure instead of other validated measures, (e.g. the numerical rating scale for breathlessness)

15. Moderate-severe cognitive difficulties was defined as ≥2 on the CPS. However 2 = mild impairment. Should this not be >2? Please clarify.

16. Please comment on missing data. How much data was missing & how was this managed?

Results

17. Is 20,773 the total number of people included, or the total number of assessments? If the former, what was the total number of assessments?

18. Results paragraph 3: ‘Patients grouped in the neurological category presented with the highest average reports on the cognitive impairment scale (91.3%).’ As in 91.3% scored ≥2 on the CPS?

19. Table 1. Do items in bold represent statistically significant differences between groups? If so, what tests were used? Please state in methods and in legend to table 1.

20. Table 1, last section: “number of assessment’s in the last 26 weeks of life”. It looks like this is actually describing the proportion of assessments that occurred at each time period within last 6 months?

21. For the trajectories, you state that all RAI-HC assessments in the last 26 weeks were used (as compared to the most recent one for the demographic info in table 1). In which case, how many assessments contributed to the trajectory analysis? I cannot see this reported – apologies if I have missed it.

22. Table 2 – “impaired cognitive performance”. Is this the same as “moderate-severe cognitive difficulties” mentioned above?

23. Table 2 – significance results are reported. What tests were used? Please add detail to methods. Also, why are some of the results with confidence intervals that don’t cross zero not highlighted as significant e.g. age >85 for moderate-severe pain= 0.51-0.69, but this is not in bold

Discussion/Conclusions

24. Para 2 “confounder”  confound

25. Please review the last two sentences of the conclusion & ensure they are linked directly to the findings. At the moment I’m struggling to see how they are based on the results of this study

Reviewer #3: The study aimed to explore symptom trajectories in non-cancer patients specifically for patients who died from four groups of conditions namely, cardiovascular, neurological, respiratory, and renal (not mutually exclusive groups).

• State the exact name of the statistical technique used in your multivariate analysis under “materials and method” in the abstract, including how the study outcomes were evaluated or coded in the multivariate model.

• State the exact P-values of the model results and the exact threshold for statistical significance used to differentiate statistically significantly from non-significant findings.

• The use of the term ‘symptom needs’ throughout the manuscript is confusing. Do you mean “symptom trajectories”? if so, change appropriately. If otherwise define what ‘symptom needs’ means in the context of your study.

• The entire method needs to be re-written and organised following appropriate reporting guidelines: see STROBE for more information. Ideally, ‘Data Sources’ ought to come before study population. https://www.strobe-statement.org/index.php?id=strobe-endorsement

• The authors should adjust for multiple comparisons (i.e. Bonferroni adjustment) and controls the familywise error rate, given the number of statistical tests conducted in the study. All results related to multivariate analyses should be re-written following adjustment for family-wise error.

• The authors should describe how the study outcomes were coded into the multivariate model in the method section. Also, no mention of P-values and level of statistical significance, including the software used to conduct statistical analysis.

• Describe the study covariates (i.e. Age, sex, marital status, and education, etc.) included in the models. Say whether it was categorical or continuous variables. If a categorical variable was used state, the levels and provide some justification for the choice of covariates used in your study.

• The information presented in Figure 1 would be better represented as a bar graph. The line graph is difficult to understand.

• Patients were grouped into four non-mutually exclusive diagnostic categories. I would argue that some patients with comorbidities would have different symptom trajectories from other patients. Therefore, the authors should account for comorbidity. Although this was mentioned as a limitation. It will be good to conduct a sensitivity analysis to explore the effect of comorbidities or perhaps adjust for this in the multivariate analysis.

6. PLOS authors have the option to publish the peer review history of their article (what does this mean?). If published, this will include your full peer review and any attached files.

Reviewer #1: No

Reviewer #2: **Yes: **Simon Etkind

Reviewer #3: No

---

## [Author Response · Author response to Decision Letter 0]

29 Apr 2021

Reviewer response

Two main issues that need to be addressed.

1. Reporting of the methods - please use reporting guidance for observational studies and extension for routine data. This is STROBE and extension RECORD. Please review these guidance, complete for your manuscript, indicate in your methods reporting using this guidance and reference, and the completed checklist included as a supplementary file.

A revised version of the methods section was generated using the STROBE and RECORD outlines as indicated. The completed checklist was included as a supplementary file. 

2. The manuscript is comparing assessments from different patients conducted at different times prior to death. This is a limitation compared to a prospective longitudinal study, as the individual trajectories cannot be seen. This should be discussed and indication of a limitation on the study. 

This is an important point. We have included a lengthy paragraph in the discussion devoted to the limitations and strengths of our approach. The paragraph reads as follows: 

“Using administrative home care data to describe the weekly average symptom prevalence in the 6 months before death has limitations and strengths. The limitations include the real potential for selection bias in that we lose out on data from patients with very complex symptom issues who then refuse home care services or when they go to hospital; thus, the symptoms of each disease group could be under-reported. We did examine those who died in hospital compared to those who died at home as a sensitivity test, and found no difference in the symptom trajectories. Also, other data shows most terminal hospitalizations are less than 2 weeks and home care is protective of end-of-life hospitalizations. Moreover, it is also possible that those with very complex symptoms would be more willing to accept home care services. Nonetheless, the timing of these formal assessments are typically far apart and only about half the patients had repeated measures, meaning that the trajectories are an average of the cohort and not individual trajectories of symptoms reported weekly. However, a strength of our approach is that it avoids some of the major issues with conducting studies at end of life, which include low recruitment, high missing data, and high drop-out rates because patients are too tired or sick to participate. Also in our study, there is virtually no missing data, as the RAI-HC is a mandatory standardized reporting tool for everyone who receives publicly-funded home care. Thus, our data is an inclusive population-based cohort, producing a large sample size, and allows us to look at trajectories over time on a weekly basis (for the subset of the cohort who reported in that week).”

 

Journal Requirements:

RESPONSE: This is done.

2) We note that you have indicated that data from this study are available upon request. PLOS only allows data to be available upon request if there are legal or ethical restrictions on sharing data publicly. For information on unacceptable data access restrictions, please see http://journals.plos.org/plosone/s/data-availability#loc-unacceptable-data-access-restrictions.

RESPONSE: We have clarified this in the “Data Availability” section in the title page. The data are not available upon request from us; they must contact Canadian Institute for Health Information as stated in the revised statement.

RESPONSE: See above. Data availability statement revised.

3) We note that you have included the phrase “data not shown” in your manuscript. Unfortunately, this does not meet our data sharing requirements. PLOS does not permit references to inaccessible data. We require that authors provide all relevant data within the paper, Supporting Information files, or in an acceptable, public repository. Please add a citation to support this phrase or upload the data that corresponds with these findings to a stable repository (such as Figshare or Dryad) and provide and URLs, DOIs, or accession numbers that may be used to access these data. Or, if the data are not a core part of the research being presented in your study, we ask that you remove the phrase that refers to these data.

RESPONSE: This statement has been removed, and all relevant data are found in the manuscript.

4) Please include your tables as part of your main manuscript and remove the individual files. Please note that supplementary tables (should remain/ be uploaded) as separate "supporting information" files.

RESPONSE: Done.

5) PLOS requires an ORCID iD for the corresponding author in Editorial Manager on papers submitted after December 6th, 2016. Please ensure that you have an ORCID iD and that it is validated in Editorial Manager. To do this, go to ‘Update my Information’ (in the upper left-hand corner of the main menu), and click on the Fetch/Validate link next to the ORCID field. This will take you to the ORCID site and allow you to create a new iD or authenticate a pre-existing iD in Editorial Manager. Please see the following video for instructions on linking an ORCID iD to your Editorial Manager account: https://www.youtube.com/watch?v=_xcclfuvtxQ

RESPONSE: Corresponding author’s ORCID ID is included.

 

Reviewers' comments: / Reviewer's Responses to Questions / Comments to the Author

1. Is the manuscript technically sound, and do the data support the conclusions?

Reviewer #1: No

Reviewer #2: Partly

Reviewer #3: No

2. Has the statistical analysis been performed appropriately and rigorously? 

Reviewer #1: Yes

Reviewer #2: Yes

Reviewer #3: No

3. Have the authors made all data underlying the findings in their manuscript fully available?

Reviewer #1: Yes

Reviewer #2: Yes

Reviewer #3: No

4. Is the manuscript presented in an intelligible fashion and written in standard English?

Reviewer #1: Yes

Reviewer #2: Yes

Reviewer #3: Yes

5. Review Comments to the Author

Reviewer #1: 

Thank you for submitting this manuscript. This large observational study on symptom trajectories of non-cancer patients in the last six months of life addresses an important topic and is generally well written. However, I have some concerns about the methodological approach and the reporting.

Major concerns:

1. The 6 month symptom trajectories are constructed from single individual symptom scores with varying time until death, rather than from individual repeated symptom measures. A potential issue with this approach is that it is very plausible that those who are close to death and more symptomatic may be more likely to decline a study visit and are missing from the sample (selection bias). This potential source of bias brings some doubt to the conclusion that symptom trajectories towards the end of life tend to be flat.

This is a very important point for discussion. We have included a lengthy paragraph describing the limitations and strengths of our approach, which include the above point of selection bias and under-reporting of symptoms. We have also revised the results and discussion to avoid the mention that the trajectories are “flat”—and we have more precisely described (using percentages) how the trends increase slightly or in some cases more dramatically over time for each symptom/disease group. Here is the revised paragraph: 

“Using administrative home care data to describe the weekly average symptom prevalence in the 6 months before death has limitations and strengths. The limitations include the real potential for selection bias in that we lose out on data from patients with very complex symptom issues who then refuse home care services or when they go to hospital; thus, the symptoms of each disease group at those points could be under-reported. We did examine those who died in hospital compared to those who died at home as a sensitivity test, and found no difference in the symptom trajectories, though those dying in hospital were more likely to have uncontrolled pain. Also, other data shows most terminal hospitalizations are less than 2 weeks and home care is protective of end-of-life hospitalizations. Moreover, it is also possible that those with very complex symptoms would be more willing to accept home care services. Nonetheless, the timing of these formal assessments are typically far apart and only about half the patients had repeated measures, meaning that the trajectories are an average of the cohort and not individual trajectories of symptoms reported weekly. However, a strength of our approach is that it avoids some of the major issues with conducting studies at end of life, which include low recruitment, high missing data, and high attrition rates because patients are too tired or sick to participate. Also in our study, there is virtually no missing data, as the RAI-HC is a mandatory standardized reporting tool for everyone who receives publicly-funded home care. Thus, our data is an inclusive population-based cohort, producing a large sample size, and allows us to look at trajectories over time on a weekly basis (for the subset of the cohort who reported in that week).”

2. The aim of the study is not clear and does not match with the study described. “Our study aimed to identify gaps in knowledge among patients with a variety of serious illnesses that would benefit from a palliative approach to care.” It is not clear to me how this study identifies knowledge gaps. Please consider what this study aims to achieve and its contribution to the literature.

We agree. We have changed this aim statement completely. It now reads: “Our study aimed to analyze the average symptom trajectories for non-cancer patients in the last six months of life and identify factors associated with having a symptom issue.”

We have also revised the introduction to more clearly explain how our study (focused on community-dwelling non-cancer patients dying at home) is distinct from prior research on non-cancer patients (mostly in hospital and institutional settings).

Minor points:

Methods

3. • Please clarify how the data is collected when the subject has cognitive impairment e.g. for self-reported health?

Good question. Certainly there would be variation by each home care assessor; but I understand that the assessor would ask the patient how they are doing, and is looking for this: “Patient feels he/she has poor health (when asked) “ (yes/no), which is the language on the tool. As such, to the reviewer’s point, the documentation of this “self-reported” measure would be expected to be much lower for those with neurological diagnoses (e.g. Alzheimer’s dementia or another type of dementia). In fact, that is what we see: that those with neurological diagnoses have lower rates of “self-reported” poor health… as they may difficulties communicating. Upon reflection, given this comment (and how this measure is different than the others, which are truly provider-reported), as well as the additional space required to discuss the limitations of our data, we have decided to eliminate this measure from the paper. The other symptoms are more significant/important to clinical care. Also, we have clarified in our methods that the RAI-HC is a provider-reported measure (and all the outcomes in the paper are provider-reported). And have used direct quotes of the language for many of the measures in the tool, to better clarify how an outcome was assessed. 

4. • “Shortness of breath was measured by asking: Do you feel short or breath, yes or no? Answers were documented based on the assessor’s judgement after the interview with the patient.” These two sentences appear contradictory. Please clarify under what circumstances the assessor would override a patient’s assessment of their shortness of breath?

We agree this is confusing and misleading. We have rewritten this to be more clear. It now reads: “Shortness of breath (item K3e): “Shortness of breath was present in the past 3 days” (yes/no)”

5. • The information provided about pain measurement is unclear. Is the main outcome the combination of moderate/severe AND uncontrolled? Please revise so that the methods are clear and reproducible.

This is correct. The main outcome is the combination of having moderate/severe pain (using the Pain scale) and the presence of “pain being uncontrolled by medication”. We have revised the methods as suggested. It now reads: “Pain: Moderate-severe daily pain that was also uncontrolled was measured by having a score of 2 or higher out of 4 on the Pain Scale (item K4a-b) (meaning the frequency is daily and the intensity is moderate to severe) and that the pain was uncontrolled (item K4e) (i.e., “medications do not adequately control pain”).”

In addition, throughout the methods, we have indicated which items from the RAI-HC were used to derive these variables (and appropriate references), which is also compliant with the RECORD/STROBE requirement. This will support the methods being clear and reproducible. 

6. • The grouping for psychosocial symptoms seems unusual - does self-reported health fit here?

As stated above, we have eliminated self-reported poor health from the paper, as it is the only one that is “self-reported”, is subject to bias in the neurological group, and is different from the other measures which are provider-reported/assessed.

7 • Please justify why you selected the 5 symptoms as the main outcomes.

We have added a justification in the methods. “Outcomes of interest were derived from the RAI-HC assessment and included pain, shortness of breath, (physical symptoms) cognitive performance and caregiver distress (psychosocial symptoms). These also correspond with two closely-related to disease groups (neurological=cognitive performance; respiratory=shortness of breath) and two general measures (pain and caregiver distress).” As stated earlier, we have eliminated the “self-reported poor health measure” from the paper.

8. • There is a wide timeframe within which RAI assessments can take place (6-12 months). Please can you detail reasons for this and the implications on the sample? For instance, could patients who are feeling more unwell decline the visits? Please can you provide some critique and discussion around the approach and possibility of bias.

We have clarified in the methods that the way the RAI-HC is administered, why it is done, and how often (at least every 6 months): “All patients expected to receive at least 60 days of home care receive a standardized assessment tool, called the RAI-HC (akin to the Minimum Data Set in the USA). This assessment tool is mandated by the province for billing, accountability, and research purposes. The RAI-HC is completed in the patient’s home by a trained professional (typically a registered nurse) on a laptop, following a detailed coding manual. Thus, the tool contains provider-reported outcome measures. The assessment is repeated at least every 6 months, unless there is a major change in health status or a discharge from hospital; thus, patients can have multiple assessments completed…” With respect to the potential for bias, as described above to comment #1, we have included a lengthy paragraph of the potential for selection bias. It is possible that patients who feel unwell could decline a nursing visit, but it is also possible patients accept the nursing visit because the nurse can help the patient feel better. It could be both. We discuss both possibilities in the new paragraph.

Results

9. • Table 1, what does bold indicate? Please detail in the footnotes.

The bolded text were errors. They have been removed.

10. • In Table 2, please detail exactly which covariates are included in each model.

We have now included all the variables in the footnotes. It reads: “Each of the four models was adjusted for these additional covariates: caregiver lives with patient; moderate-severe impairment in Activities of Daily Living; social decline causing distress; signs and symptoms of depression; and loss of appetite.”

Discussion:

11. • In discussion: “Based on our descriptive and multivariate analyses, we could demonstrate that patients with non-cancer overall seem not to suffer severely from symptom needs in their disease trajectory over the last six months of life.” What is this based on? I don’t think I would conclude the same based on the data presented in the figures.

Thank you for this comment. We agree that this statement does not fit our findings. It has been removed. We have greatly revised the manuscript. This paragraph in the discussion has been changed to summarize our findings in the results as such: “Our data present trajectories of symptoms in the last six months of life in a non-cancer population of home care patients among four disease groups: cardiovascular, neurological, renal, and respiratory. Across all non-cancer disease groups, the trajectory of symptom prevalence increased slightly each week towards death. Cognitive impairment was evident in at least half of the patients in the disease groups, and over 90% in the neurological group. Prevalence of shortness of breath rose by 20% over time across all groups, with the highest prevalence being among those with respiratory disease at 86% in the last week of life. Caregiver distress rose by 10% over time and was prevalent in 35%-40% of patients in the final weeks of life. With a sample size of 20,773 assessments, this is a very large population-based cohort focusing on describing average weekly symptom prevalence among those dying at home.”

12. • Hypothesis referred to in discussion does not correspond to the hypothesis detailed in the background. Please address this discrepancy.

Based on other comments, we have eliminated the hypothesis from our introduction (and thus our discussion).

13. • The trajectories presented are hypothetical and do not relate to individual trajectories. Please discuss the possible implications and the limitations of this approach.

As discussed in response to comment #1, we have included a lengthy paragraph exploring the limitations and strengths of our “average” trajectory approach. 

14. • Overall the manuscript would benefit from adhering to reporting guidelines, e.g. STROBE for observational studies.

Thank you. We have revised our methods and the entire paper to comply with the STROBE and RECORD outlines as indicated. The completed checklist was included as a supplementary file.  

Reviewer #2: 

Thanks for the opportunity to review this manuscript, which uses a robust regional routinely collected clinical dataset to retrospectively investigate symptom trajectories of people who died from non-cancer illnesses. I think the approach taken is appropriate, however there are several areas in which the reporting of the methods could be improved, or clarification is needed, therefore I think revisions are required before publication. I have made comments below to suggest how this might be done.

General comments

1. I think the methods section could be organised more clearly to aid the reader (see specific comments below.) I would also recommend that the authors use a checklist for reporting of this type of study e.g. the RECORD statement

https://journals.plos.org/plosmedicine/article?id=10.1371/journal.pmed.1001885

Thank you. We revised the methods and the manuscript to comply with the STROBE and RECORD outlines as indicated. The completed checklist was included as a supplementary file. 

2. Focusing on deaths at home/in hospital only means that the study is unable to comment on those who died in a care home or hospice (what proportion of deaths is this in Ontario?). These two groups might have very different symptom burden and so this is a limitation of the study which should be discussed.

We agree. Data indicates that about 3% of patients die in hospices in Ontario (although the vast majority use home care services prior to death, and the median time in hospice in Ontario is 18 days). Approximately 25% die in care homes (aka long-term care or nursing homes). Although the latter would likely not be in our sample, as the average time in a long-term care home is 18 months before death. We have included this in our limitations sections. It reads: “Focusing on users of publicly-funded home care at the end of life means we do not have data on those who did not use home care services or strictly used private home care services or died in long-term care (approximately 20-25% of the population).”

3. As I understand it the study is looking at ‘average trajectories’ i.e. rather than looking at the change in each individual’s symptoms over 6 months, the authors are comparing assessments from different patients conducted at different times prior to death. This is a limitation compared to a prospective longitudinal study, as the individual trajectories cannot be seen. This should be discussed.

We are indeed describing average trajectories. As the reviewer correctly states, there are advantages and disadvantages to this, which we now discuss further in our discussion section in its own paragraph as follows:

“Using administrative home care data to describe the weekly average symptom prevalence in the 6 months before death has limitations and strengths. The limitations include the real potential for selection bias in that we lose out on data from patients with very complex symptom issues who then refuse home care services or when they go to hospital; thus, the symptoms of each disease group at those points could be under-reported. We did examine those who died in hospital compared to those who died at home as a sensitivity test, and found no difference in the symptom trajectories, though those dying in hospital were more likely to have uncontrolled pain. Also, other data shows most terminal hospitalizations are less than 2 weeks and home care is protective of end-of-life hospitalizations. Moreover, it is also possible that those with very complex symptoms would be more willing to accept home care services. Nonetheless, the timing of these formal assessments are typically far apart and only about half the patients had repeated measures, meaning that the trajectories are an average of the cohort and not individual trajectories of symptoms reported weekly. However, a strength of our approach is that it avoids some of the major issues with conducting studies at end of life, which include low recruitment, high missing data, and high attrition rates because patients are too tired or sick to participate. Also in our study, there is virtually no missing data, as the RAI-HC is a mandatory standardized reporting tool for everyone who receives publicly-funded home care. Thus, our data is an inclusive population-based cohort, producing a large sample size, and allows us to look at trajectories over time on a weekly basis (for the subset of the cohort who reported in that week).”

4. Given the aim, I'm not sure how the multivariable analysis adds to this study. It doesn’t add to the analysis of symptom trajectories, since it only gives the odds of having each outcome at any point in the last 6 months of life dependent on the characteristic, not the change over time. Please could the authors more clearly justify why they conducted this analysis and how it contributes to their aim.

The aim was to describe the trajectory of physical and psychosocial symptoms for non-cancer patients (in a home care population) in the last 6 months of life. The results of this show that the prevalence of the symptoms are evident in half or so (sometimes higher or lower), depending on the symptom. To us, this begs the question of the factors associated with having the symptom at all during the last 6 months compared to those who don’t have the symptom. That is why we did the regression. You are correct that it doesn’t show the change, or the incidence of having the symptoms, which is a different research question (albeit an important one—and the sample would drop dramatically for those without a repeat measure in our study window). We believe having a regression showing the factors associated with the prevalence of having the symptom in the last 6 months is useful. We have revised the aim to read: “Our study aimed to describe the average symptom trajectories for a cohort of non-cancer patients in the last six months of life and identify factors associated with having a symptom issue.”

Specific comments

Abstract

5. Please state the data source explicitly in the abstract. E.g. ‘retrospective study using data from the Canadian institute for health.’

We have revised the abstract to explicitly name the data sources used.

6. The aim in the abstract does not match that in the main paper. I think that the aim is probably to analyse symptom trajectories and identify differences, rather than specifically to identify gaps in knowledge? Please could you clarify and ensure consistency between abstract and main paper.

We agree. We have revised our aim. In the manuscript, it now reads: “Our study aimed to describe the average symptom trajectories for a cohort of non-cancer patients in the last six months of life and identify factors associated with having a symptom issue.” It has been revised in the abstract as well. 

7. “Patients were grouped into four non-cancer disease groups such as”. There is no need for “such as”, all groups are described

We have made this change.

8. When reporting odds in the abstract, please state the comparator. E.g. 'renal patients had higher odds of pain compared to other groups' etc.

This has been corrected. A comparator group has been included in all relevant sentences.

9. “symptom trajectories vary with disease group”. Do the trajectories differ or is it symptom prevalence that varies?

We have revised this sentence to be more clear. It now reads: “In our cohort of non-cancer patients dying in the community with home care services, the trajectory of symptom prevalence increased over time across all disease groups.”

Introduction

10. Introduction line 4 “satisfactory” – do you mean satisfaction?

We have edited this typo/grammatical mistake. It has been changed to “satisfaction with care”

11. 3rd sentence. How does the possibility that palliative care referrals are often made for symptom management explain the reduced referrals in non-cancer diagnoses? Symptoms are known to be high in non-cancer too (as the authors discuss later on). Please rephrase to clarify the argument.

We agree that this is confusing. We have heavily revised the entire introduction to be more clear about the problem, the prior research and gaps, and how our study aims to address that gap. 

12. Please move the description of the frequency with which the RAI-HC is completed to the methods section.

This has been done. Complying with the STROBE/RECORD format, the explanation of RAI-HC is found in the “data sources” section.

13. I think the aim in the introduction is clear, but I’m a bit confused by the hypothesis: why do the authors hypothesise different symptom patterns in different non cancer illnesses? What existing evidence has led them to this hypothesis? Also, by “gaps of knowledge” do the authors mean differences in symptom patterns which would therefore allow a more nuanced approach to palliative care referral? Please clarify.

Thank you for this comment. Upon reflection, the hypothesis was confusing and did not add to the paper. We have removed the hypothesis. Our revisions to the introduction and discussion have hopefully made the paper tighter and clearer.

Methods

14. At the start of the methods, please state that you are using routinely collected clinical data.

This is done. We have included this statement as the first sentence under the “data sources” section: “We used routinely collected clinical health administrative data.”

15. The RAI-HC may not be familiar to international readers. It would be helpful if it could be introduced and described in a single section. At the moment the description is spread across the introduction and several sections of the methods. Perhaps this information can be combined into a single description of what the RAI-HC is, how it is completed & how it was used here

As suggested, the description of RAI-HC is now combined into a single section, under the “data sources” section, which also complies with RECORD/STROBE.

16. Re: diagnostic categories, were there no deaths with liver failure? Or were these combined into another category

Good question. We reviewed the list of all diagnosis. As a tool, not only for end of life, but any home care use, the other categories include things like osteoporosis, hip fracture, asthma or head trauma. There are 28 options, but none are for liver failure. The next question is an option to add in other diagnosis and their relevant ICD-10 code, where liver failure might have been listed, but we did not explore this additional item. 

17. Last sentence of ‘population’ section. I think this would fit better at the start of the 'analysis' section.

We agree. As suggested, this sentence has been moved to the “statistical methods” section.

18. Whilst the pain outcome is detailed, the shortness of breath outcome is a yes/no question. I recognise the authors are limited by the dataset, but could they comment on the effect on symptom prevalence of using this measure instead of other validated measures, (e.g. the numerical rating scale for breathlessness)

We have added this as a limitation. We agree, unfortunately, this measure is only dichotomous, and is not categorical or more sophisticated/detailed such as other valid measures. We have included a systematic review of dyspnea measures as a citation here.

19. Moderate-severe cognitive difficulties was defined as ≥2 on the CPS. However 2 = mild impairment. Should this not be >2? Please clarify.

Thank you for catching this error. Our definition was >=2, so we have corrected our description to mild to severe cognitive impairment. 

20. Please comment on missing data. How much data was missing & how was this managed?

This has been included/addressed in the paragraph about the strengths and limitations of using admin data for this study, and describing the average overall symptom trajectory vs. individual symptom scores. The summary of that is that because it’s administrative data, completing the RAI-HC is mandatory for reporting, and there is virtually no missing data. Every field/item is completed. So imputation or other methods were not required. This is a major strength of using this dataset.

Results

21. Is 20,773 the total number of people included, or the total number of assessments? If the former, what was the total number of assessments?

We have clarified this in our results. It now reads: “…The final sample size was 20,773 unique individuals (33,596 assessments).” [in the final 6 months of life]

22. Results paragraph 3: ‘Patients grouped in the neurological category presented with the highest average reports on the cognitive impairment scale (91.3%).’ As in 91.3% scored ≥2 on the CPS?

This is correct. Of those groups in the neurological disease group (i.e. had Alzheimer’s dementia, dementia (other than Alzheimer’s), multiple sclerosis, parkinsonism), 91.3% had mild-severe cognitive impairment (as per the CPS—i.e. a score of 2 or more on the CPS) in their assessment closest to death. We have revised this sentence to be more clear. This paragraph now reads: “Examining outcomes in the last assessment closest to death, there was a higher prevalence of moderate-severe pain in the cardiovascular (57.2%), renal (61.0%) and respiratory group (58.3%), compared to the neurological group (42.7%) (Table 1). 91.3% of patients with neurological disease had documented mild-severe cognitive impairment.” 

23. Table 1. Do items in bold represent statistically significant differences between groups? If so, what tests were used? Please state in methods and in legend to table 1.

This was an error. Table 1 no longer has any items in bold. 

24. Table 1, last section: “number of assessment’s in the last 26 weeks of life”. It looks like this is actually describing the proportion of assessments that occurred at each time period within last 6 months?

This was an error. We have clarified this in the Table 1.It now reads: “Timing of patient’s closest assessment to death” [where n (%) are correct]

25. For the trajectories, you state that all RAI-HC assessments in the last 26 weeks were used (as compared to the most recent one for the demographic info in table 1). In which case, how many assessments contributed to the trajectory analysis? I cannot see this reported – apologies if I have missed it.

We have clarified this in our results and included this information. It now reads: “…The final sample size was 20,773 unique individuals (33,596 assessments).” [in the final 6 months of life]. So there were 33,596 assessments making up the trajectory graphs.

27. Table 2 – “impaired cognitive performance”. Is this the same as “moderate-severe cognitive difficulties” mentioned above?

We have changed Table 2 heading to “mild-severe cognitive impairment” for clarify and consistency. 

28. Table 2 – significance results are reported. What tests were used? Please add detail to methods. Also, why are some of the results with confidence intervals that don’t cross zero not highlighted as significant (e.g. age >85 for moderate-severe pain= 0.51-0.69), but this is not in bold

The tests of significance are now mentioned in the methods.

All significant findings have now been bolded. (with an indication in the footnotes)

Discussion/Conclusions

28. Para 2 “confounder”  confound

This was corrected. 

29. Please review the last two sentences of the conclusion & ensure they are linked directly to the findings. At the moment I’m struggling to see how they are based on the results of this study

Thank you. Our entire conclusion has been revised to closely fit the results of our study. It now reads: “In conclusion, our study describes symptom trajectories in non-cancer home care recipients in Ontario, Canada at end of life. We found across all non-cancer disease groups, the trajectory of symptom prevalence increased slightly each week towards death. Moderate to severe pain was prevalent in nearly half or more of the cohort, but only one-fifth described the pain as uncontrolled. In contrast, shortness of breath, impaired cognitive function and caregiver distress were more highly and consistently prevalent across time near the end of life. Our results suggest the non-cancer population has unmet symptoms needs outside institutional settings.” 

 

Reviewer #3: 

The study aimed to explore symptom trajectories in non-cancer patients specifically for patients who died from four groups of conditions namely, cardiovascular, neurological, respiratory, and renal (not mutually exclusive groups).

1 • State the exact name of the statistical technique used in your multivariate analysis under “materials and method” in the abstract, including how the study outcomes were evaluated or coded in the multivariate model.

We have included the name of the regression (multivariate logistic regression), including definitions of our study outcomes in the methods of our abstract.

2 • State the exact P-values of the model results and the exact threshold for statistical significance used to differentiate statistically significantly from non-significant findings.

We have included the p-value threshold for statistical significance in the methods and the Figure.

3 • The use of the term ‘symptom needs’ throughout the manuscript is confusing. Do you mean “ symptom trajectories”? if so, change appropriately. If otherwise define what ‘symptom needs’ means in the context of your study.

We have remove the term “symptom needs” from the paper. We have replaced it with “symptom prevalence” for clarity.

4 • The entire method needs to be re-written and organised following appropriate reporting guidelines: see STROBE for more information. Ideally, ‘Data Sources’ ought to come before study population. https://www.strobe-statement.org/index.php?id=strobe-endorsement

A revised version of the methods section was generated using the STROBE and RECORD outlines as indicated. The completed checklist was included as a supplementary file. We followed the STROBE/RECORD guidelines in terms of order of information presented.

5 • The authors should adjust for multiple comparisons (i.e. Bonferroni adjustment) and controls the familywise error rate, given the number of statistical tests conducted in the study. All results related to multivariate analyses should be re-written following adjustment for family-wise error.

The issue of multiple testing is an important one. We were concerned about using Bonferroni (or other correction methods) because several papers on this topic concluded that the these test are not helpful in interpreting results, and can lead to the p-value being too stringent, leading to higher rates of Type 2 error. See for example (i) TV Perneger. What’s wrong with Bonferroni adjustments. BMJ. 1998 Apr 18; 316(7139):1236-1238; and also (ii) P Ranganathan, CS Pramesh, M Buyse. Common pitfalls in statistical analysis: The perils of multiple testing. Perspectives in Clinical Research. 2016 Apr-Jun; 7(2):106-107) amongst others. As suggested in these papers, we used a pragmatic approach of describing the statistical tests performed, and interpreting results cautiously, understanding the wider context of observed data with the number of tests performed, so as to provide a more balanced ability to interpret results.

6 • The authors should describe how the study outcomes were coded into the multivariate model in the method section. Also, no mention of P-values and level of statistical significance, including the software used to conduct statistical analysis.

We have provided more details about the exact definitions of how each outcome and covariates were coded in the multivariate model in the method’s section (including the items # of the tool). We have included mention of p-values and level of statistical significance (p<0.05). And we have included the software used to conduct the analysis (SAS v. 9.4)

7 • Describe the study covariates (i.e. Age, sex, marital status, and education, etc.) included in the models. Say whether it was categorical or continuous variables. If a categorical variable was used state, the levels and provide some justification for the choice of covariates used in your study.

We have more clearly described each study covariates that were included in the regression models, which are now described in a sub-section titled, ‘Covariates’, in the Methods section. We describe and define them clearly as dichotomous (non were categorical). We have included a reference for justification for these variables. 

8 • The information presented in Figure 1 would be better represented as a bar graph. The line graph is difficult to understand.

Because we have 4 disease groups, and we have data for each group at each of the 26 weeks before death, if we created a bar graph… we would have 4 bars for each week before death… which would be 104 bars on the graph. We are concerned this would be overwhelming for the reader. Given we are trying to describe the trajectory of symptom profile over time, we feel a line graph, one line per disease group, would better reflect our main aim. We have however, modified our figures so that the graphs are bigger and the legends/text outside take up less space. We hope these formatting changes make the graph easier to understand. 

9 • Patients were grouped into four non-mutually exclusive diagnostic categories. I would argue that some patients with comorbidities would have different symptom trajectories from other patients. Therefore, the authors should account for comorbidity. Although this was mentioned as a limitation. It will be good to conduct a sensitivity analysis to explore the effect of comorbidities or perhaps adjust for this in the multivariate analysis.

Unfortunately, our data were not designed for controlling complex comorbidities. Thus, while it can identify if a disease was present, e.g., cardiovascular disease as being the main reason for visit, it is not comprehensive in detailing all the comorbidities of the patient. For instance, we are not able to look at all the physician and hospital billings for the past 2 years of the patient’s care which would be a more accurate way to measure comorbidities (this is how they would do it using Deyo-Charlson comorbidity index or Elixhauser score). Therefore, we are not confident that the home care data properly measures comorbidities, though it would detail the main reason for home care admission. We are also not able to determine the severity of the comorbidity. We do include ADL self-performance hierarchy scale, which does take into account people’s ability to do activities of daily living—which is a partial measure of how comorbidities would affect daily life, but that is not a specific measure of the number or severity of additional comorbidities. We did further describe in our limitations that we “are unable to control for specific comorbidities.” This analysis is out of scope with our data but is a good question that would require more data linkage and a more in-depth study, which we have mentioned as future research.

 

6. PLOS authors have the option to publish the peer review history of their article (what does this mean?). If published, this will include your full peer review and any attached files.

Do you want your identity to be public for this peer review? For information about this choice, including consent withdrawal, please see our Privacy Policy.

Reviewer #1: No

Reviewer #2: Yes: Simon Etkind

Reviewer #3: No

1. Seow H, Qureshi D, Isenberg SR, Tanuseputro P. Access to Palliative Care during a Terminal Hospitalization. J Palliat Med. 2020. Epub 2020/02/06. doi: 10.1089/jpm.2019.0416. PubMed PMID: 32023424.

---

## [Decision Letter · Decision Letter 1]

24 May 2021

Symptom trajectories of non-cancer patients in the last six months of life: Identifying needs in a population-based home care cohort

PONE-D-21-00638R1

Dear Dr. Conen,

We’re pleased to inform you that your manuscript has been judged scientifically suitable for publication and will be formally accepted for publication once it meets all outstanding technical requirements.

Kind regards,

Catherine J Evans, PhD, MSc, BSc (Hons)

Academic Editor

PLOS ONE

Additional Editor Comments (optional):

Your careful and considered responses to the comments from the peer reviewers has greatly improved the calibre of the reporting in this manuscript. I am pleased to accept the manuscript for publication. Please complete a final proof of the manuscript and address the comments by #Reviewer 1 re revising the following sentence to improve clarity as the point it is making is unclear and generally confusing (under ‘Outcomes’ in methods section):

These also correspond with two closely related to disease groups (neurological=cognitive performance; respiratory=shortness of breath) and two general measures (pain and caregiver distress). responded carefully to the editor and peer reviewer comments.

Reviewers' comments:

Reviewer's Responses to Questions

**Comments to the Author**

1. If the authors have adequately addressed your comments raised in a previous round of review and you feel that this manuscript is now acceptable for publication, you may indicate that here to bypass the “Comments to the Author” section, enter your conflict of interest statement in the “Confidential to Editor” section, and submit your "Accept" recommendation.

Reviewer #1: All comments have been addressed

Reviewer #3: All comments have been addressed

2. Is the manuscript technically sound, and do the data support the conclusions?

Reviewer #1: Yes

Reviewer #3: Yes

3. Has the statistical analysis been performed appropriately and rigorously? 

Reviewer #1: Yes

Reviewer #3: Yes

4. Have the authors made all data underlying the findings in their manuscript fully available?

Reviewer #1: Yes

Reviewer #3: (No Response)

5. Is the manuscript presented in an intelligible fashion and written in standard English?

Reviewer #1: Yes

Reviewer #3: Yes

6. Review Comments to the Author

Reviewer #1: This manuscript is much improved. Please consider revising the following sentence as the point it is making is unclear and generally confusing (under ‘Outcomes’ in methods section):

These also correspond with two closely related to disease groups (neurological=cognitive performance; respiratory=shortness of breath) and two general measures (pain and caregiver distress).

Reviewer #3: (No Response)

7. PLOS authors have the option to publish the peer review history of their article (what does this mean?). If published, this will include your full peer review and any attached files.

Reviewer #1: No

Reviewer #3: No

---

## [Editor Report · Acceptance letter]

3 Jun 2021

PONE-D-21-00638R1 

Symptom trajectories of non-cancer patients in the last six months of life: Identifying needs in a population-based home care cohort 

Dear Dr. Conen:

I'm pleased to inform you that your manuscript has been deemed suitable for publication in PLOS ONE. Congratulations! Your manuscript is now with our production department. 

Kind regards, 

on behalf of

Dr. Catherine J Evans 

Academic Editor

PLOS ONE